# Mitochondrion-specific dendritic lipopeptide liposomes for targeted sub-cellular delivery

Lei Jiang [1], Sensen Zhou[1], Xiaoke Zhang[1], Cheng Li[1], Shilu Ji[1], Hui Mao [2] & Xiqun Jiang [1]✉

The mitochondrion is an important sub-cellular organelle responsible for the cellular energetic source and processes. Owing to its unique sensitivity to heat and reactive oxygen species, the mitochondrion is an appropriate target for photothermal and photodynamic treatment for cancer. However, targeted delivery of therapeutics to mitochondria remains a great challenge due to their location in the sub-cellular compartment and complexity of the intracellular environment. Herein, we report a class of the mitochondrion-targeted liposomal delivery platform consisting of a guanidinium-based dendritic peptide moiety mimicking mitochondrion protein transmembrane signaling to exert mitochondrion-targeted delivery with pH sensitive and charge-reversible functions to enhance tumor accumulation and cell penetration. Compared to the current triphenylphosphonium (TPP)-based mitochondrion targeting system, this dendritic lipopeptide (DLP) liposomal delivery platform exhibits about 3.7-fold higher mitochondrion-targeted delivery efficacy. Complete tumor eradication is demonstrated in mice bearing 4T1 mammary tumors after combined photothermal and photodynamic therapies delivered by the reported DLP platform.

[1] MOE Key Laboratory of High Performance Polymer Materials and Technology, and Department of Polymer Science and Engineering, College of Chemistry and Chemical Engineering, Nanjing University, Nanjing, China. [2] Department of Radiology and Imaging Sciences, Emory University, Atlanta, GA, USA. ✉email: jiangx@nju.edu.cn

Mitochondrion not only is the energy source of the cellular functions but also plays many regulatory roles in cell apoptosis[1,2]. Mitochondrial dysfunction may lead to the initiation and progression of many diseases, including cancers, cardiovascular, neurodegenerative, and metabolic diseases, as well as aging[3,4]. Therefore, targeting mitochondria to alter and treat diseases has long been sought after[5–8]. Recently, increasing effort has been made to develop mitochondrion-targeted therapy, especially photothermal therapy (PTT) and photodynamic therapy (PDT) for treating cancer, as mitochondria are sensitive to heat shock[9,10]. With the oxygen-rich cellular environment for producing reactive oxygen species (ROS), mitochondria may be the most suitable organelle for PDT to trigger cell apoptosis[11–13]. Thus, targeted delivery of PTT and PDT agents to the mitochondria of cancer cells is highly desirable for effectively treating cancer while limiting the side effects to healthy tissues. Owing to the very short lifetime (∼40 ns) and diffusion radius (∼20 nm)[14], the cytotoxic ROS induced by PDT need to be generated in the vicinity of targeted organelle or cellular compartments in order to be effective. However, specific delivery of therapeutics to a sub-cellular component is a great challenge due to the presence of various physiological and biological barriers.

Currently, delocalized lipophilic cations (e.g., triphenylphosphonium, TPP) are commonly used as mitochondrion-targeted agents to deliver the bioactive cargo into mitochondria[15–17]. However, the results are dismal due to low targeting efficiency, low tumor accumulation, and systemic toxicity[5,18]. Thus, efficient and safe mitochondrion-targeted delivery systems are yet to be developed. It is noteworthy that most mitochondrial proteins are synthesized on cytosolic ribosomes and then relocated into mitochondria for subsequent functions. This transportation is dependent on mitochondrial precursor proteins (MPP), which generate targeting signals that direct proteins trafficking into mitochondria[19–21]. Importantly, many known MPPs share two common characteristics required for successfully directing proteins into mitochondria[22,23], an essential homologous amino acid sequence (including arginine) and amphiphilic N-terminal regions. Thus, designing a molecule with a specific amino acid moiety to mimic the pivotal sequence responsible for signaling a therapeutic agent to enter mitochondria is a logical strategy to effectively exert mitochondrion-targeted delivery.

Here, we report a rational design and implementation of a mitochondrion-targeted multifunctional nanoplatform based dendritic lipopeptide (DLP) modified with arginine residues, using the dendritic peptide as a signaling and guiding group to enable the sub-cellular delivery. For mimicking the basic amino acid domains in natural MPP, dendritic arginine-rich architecture is designed to amplify interactions of the delivery system with mitochondria using peripheral multivalency of dendrimers[24–26]. On the other hand, the stearoyl group was chosen as a hydrophobic segment to increase the lipophilicity that can promote the affinity between amphiphilic carriers and mitochondria membrane[27,28]. To maximize the tumor accumulation and minimize the normal tissue uptake, an acid-cleavable 2,3-dimethylmaleic acid (DA) which could be removed from DLP in an acidic tumor microenvironment was incorporated on the DLP to block the cationic property of arginine. Finally, the mitochondrion-targeted liposome (L-G2R-DA) was assembled with five components, i.e., dendritic lipopeptides, soy phosphatidylcholine (SPC), cholesterol, polyethylene glycol-distearoyl phosphatidylethanolamine (DSPE-PEG$_{2000}$), and photosensitizer indocyanine green (ICG) as illustrated in Fig. 1. The specificity and efficiency of mitochondrion-targeting of the reported dendritic amino acid-based liposomes were compared with those of TPP cation-based liposomes. The efficacies of ICG-loaded L-

G2R-DA in treating cancer using PTT and PDT were systematically evaluated both in vitro and in vivo. The results showed a 3.7-fold increase in mitochondria targeting by the reported L-G2R-DA over TPP cation-based liposomes and completely eradicated tumors in the 4T1 breast cancer mouse model.

## Results

**The efficiency and improvement of mitochondrion targeting.** The first-generation of arginine containing dendritic lipopeptide (G1R), second-generation of arginine containing dendritic lipopeptide (G2R), and second-generation lysine containing dendritic lipopeptide (G2K) were first synthesized (Supplementary Scheme 1–3 and Supplementary Figs. 1–6). Subsequently, G2R was decorated with an acid-cleavable 2,3-dimethylmaleic acid (DA) to generate G2R-DA (Supplementary Scheme 4 and Supplementary Figs. 7–8) according to the previous work[6,29]. For comparison studies, TPP-functionalized DSPE (DTPP) (Supplementary Scheme 5 and Supplementary Figs. 9–10) and G2R-SA, i.e., G2R decorated with an acid-non-cleavable succinic anhydride (SA) (Supplementary Scheme 6 and Supplementary Figs. 11–12), and G2K-DA (Supplementary Scheme 7 and Supplementary Figs. 13–14) were prepared as controls.

Next G1R, G2R, G2K, DTPP, G2R-SA, G2R-DA, and G2K-DA were assembled with SPC, cholesterol, DSPE-PEG$_{2000}$ using the classical thin-film hydration method to form different derivatives of liposomes named L-G1R, L-G2R, L-G2K, L-TPP, L-G2R-SA, L-G2R-DA, and L-G2K-DA (Fig. 2a), respectively. Except for G1R and DTPP with a double addition in L-G1R and L-TPP, all molar ratios of components in the liposomes were kept the same. The average sizes and zeta potentials of the above liposomes were measured by dynamic light scattering (DLS). The average sizes of liposomes were about 130, 180, 190, 140, 105, 110, and 104 nm for L-G1R, L-G2R, L-G2K, L-TPP, L-G2R-SA, L-G2R-DA, and L-G2K-DA (Fig. 2b and Supplementary Table 1), respectively. The zeta potentials of L-G1R, L-G2R, L-G2K, and L-TPP were positive and in the range of 7–24 mV. While, the zeta potentials of L-G2R-SA, L-G2R-DA, and L-G2K-DA were negative at about −15 mV (Fig. 2c and Supplementary Table 1).

The mitochondrion targeting properties of various liposomes were then evaluated in vitro. Liposomes were labeled with a red fluorescent dye (DiI) while mitochondria were labeled with a green fluorescent dye (Mitotracker Green FM). Measurement of co-localized fluorescent signals (yellow) from the liposomes and mitochondria (Fig. 2d and Supplementary Fig. 15) by confocal laser scanning microscopy (CLSM) revealed that mitochondria-targeting efficiencies of L-G2R and L-G2R-DA were substantially higher than TPP decorated liposomes (L-TPP). L-G2K also exhibited targeted accumulation in mitochondria, as the mitochondrion-localized signal of L-G2K was stronger than that of L-TPP but weaker than that of L-G2R and L-G2R-DA. When using flow cytometry to quantify the number of different liposomes in the mitochondria isolated from 4T1 breast cancer cells, L-G2R presented approximatively 5.7, 6.3, and 4.2-fold higher accumulation in mitochondria than L-TPP, L-G1R, and L-G2K, respectively (Fig. 2e). These results indicate that the reported second-generation dendritic arginine-rich structure of G2R significantly improves the mitochondrion-targeting ability with much higher targeting efficiency than those of the first-generation arginine-modified liposomes L-G1R, second-generation dendritic lysine-rich liposomes L-G2K and TPP-decorated liposomes L-TPP. Specifically, the amount of L-G2R-DA in the mitochondria of 4T1 cells was approximately 3.7-fold higher than that of TPP decorated liposomes (Fig. 2e). L-G2K-DA also exhibited well-targeted accumulation in mitochondria (Supplementary Fig. 15), but the amount in mitochondria was only one-fourth of that of L-G2R-DA.

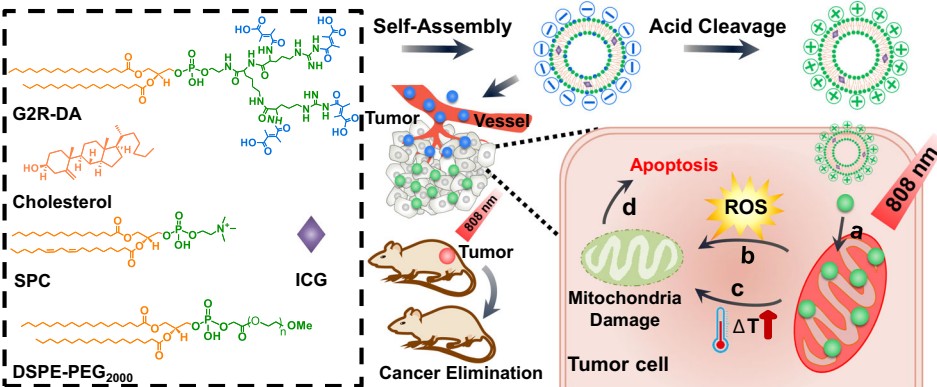

**Fig. 1 Schematic illustration of the mitochondrion-specific dendritic lipopeptide liposomes L-G2R-DA for cancer therapy. a** Mitochondrial targeting. **b** ROS production. **c** PTT-induced hyperthermia. **d** Apoptosis induced by mitochondria-targeted treatment.

**Possible mechanism of mitochondrion targeting**. To elucidate the mechanism of excellent mitochondria-targeting by the reported liposomes, we initially used various blockers of specific cellular endocytosis pathways to interrogate the processes of L-G2R internalization in 4T1 cells. As shown in Fig. 3a, the cellular uptake of L-G2R decreased remarkably in the presence of cytochalasin B ($p < 0.01$), an inhibitor of macropinocytosis.

As shown in the images from CLSM (Fig. 3b), the fluorescent signals of lysotracker were merged with L-G2R loaded dye coumarin (C6) after incubation for 1 h, indicating that L-G2R were trapped in the endosomes/lysosomes. Four hours later, expanded distribution of signals from C6-loaded L-G2R showed in the cells was observed, indicating L-G2R could mediated endosome escape for cytoplasmic liberation as illustrated in Fig. 3c. Further, we measured proteins adsorbed on L-G2R using mass spectrometry and analyzed the bound proteins on liposomes according to Cellular Component (CC) using GeneOntology (GO). Compared to the materials in the control group (liposomes without the G2R component, Supplementary Fig. 16), those in the L-G2R group adsorbed the unique mitochondrial proteins, e.g., mitochondrial inner membrane presequence translocases complex (TIM23 complex) (Fig. 3d). The TIM23 translocase consists of the membrane proteins Tim23, Tim17, and Tim50. In addition, the components of the translocase of the outer membrane (TOM complex), Tom70, Tom22, and Tom40, were also found in the mass spectrometry analysis. TOM complex is generally considered as the main gate for molecules entering into mitochondria. Thus, our observations suggest that L-G2R may directly across the TOM and TIM23 machinery into the mitochondrial matrix, which is the same as a typical precursor protein with an amino-terminal presequence[20,30] (Fig. 3e). Importantly, transmission electron microscopy (TEM) showed that L-G2R loaded with colloidal gold was located in mitochondria matrix (Supplementary Fig. 17), providing further evidence that L-G2R is able to deliver payloads into the mitochondria matrix. Thus, a possible mechanism of mitochondrion targeting by L-G2R can be considered as: (1) L-G2R liposomes enter cells through macropinocytosis and then undergo endosome escape to the cytoplasm; (2) L-G2R liposomes in the cytoplasm are transported into the mitochondrial matrix by the TOM and TIM23 mediated pathway due to the selective adsorption of mitochondrial membrane presequence translocases compared to liposomes without the G2R component.

**Preparation of ICG-loaded liposomes**. Photosensitizer ICG was successfully encapsulated in L-G2R-DA (ICG/L-G2R-DA), L-G2R-SA (ICG/L-G2R-SA), and L-TPP (ICG/L-TPP) for the ICG induced PTT and PDT. A significant redshift of ICG absorption

peak in the UV absorption spectra of ICG-loaded liposomes compared to free ICG (Fig. 4a), suggested that ICG was successfully incorporated into different formulations of the liposomes. The detailed characterizations of ICG-loaded liposomes, including size, zeta potential, and ICG encapsulation efficiency, were summarized in Supplementary Table 1. Specifically, the average sizes of ICG/L-G2R-DA, ICG/L-G2R-SA, and ICG/L-TPP in the medium at pH 7.4 as measured by DLS were $106.7 \pm 1.8$, $108.4 \pm 1.6$, and $139.4 \pm 2.5$ nm (Fig. 4b and Supplementary Table 1). The Cyro-transmission electron microscopy (Cyro-TEM) images revealed that ICG/L-G2R-DA, ICG/L-G2R-SA, and ICG/L-TPP liposomes were spherical vesicles with an average diameter of ~100 nm (Fig. 4c and Supplementary Fig. 18). The thickness of these liposomal membranes was estimated at about 5 nm, which was in accordance with the bilayer thickness of liposomes reported in the previously worked liposomes[31,32]. Next, we investigated the pH-responsive cleavage of DA groups from ICG/L-G2R-DA by measuring the pH-dependent change of zeta potential of samples. At normal physiological pH of 7.4, the zeta potential of ICG/L-G2R-DA was negative ($-15$ mV). However, it became positive (4.65 mV) at pH 6.8 and further elevated to 13.0 mV at pH 5.5 (Fig. 4d). Observed pH-dependent zeta potential change is likely attributed to arginine residues exposed on the surface after the DA group was removed under the acidic condition as schematically illustrated in Fig. 4e. Furthermore, the zeta potential of ICG/L-G2R-DA was gradually elevated within 12 h in the medium with pH of 6.8 and 5.5 (Supplementary Fig. 19). In contrast, there was no significant change in the surface zeta potential of ICG/L-TPP and ICG/L-G2R-SA in the pH range of 7.4–5.5 over 12 h (Supplementary Fig. 20). It is worth noting that there was no DLS measurable change in the sizes of ICG/L-G2R-DA at pH of 7.4, 6.8, and 5.5 (Fig. 4b). When evaluating the photothermal effect of ICG/L-G2R-DA, we observed a concentration-dependent temperature increase under irradiation of the 808 nm laser (Fig. 4f). Moreover, the temperature increase by ICG/L-G2R-DA was much higher than free ICG at the same concentration, indicating that encapsulated ICG has an enhanced photothermal effect compared to free ICG.

**Apoptosis induced by irradiation**. PDT-induced ROS generated in 4T1 cells by the various ICG-loaded liposomes after 808 nm laser irradiation ($0.65 \text{ W/cm}^2$) was measured using a ROS-sensitive green fluorescent probe, 2,7-duchlorodihydrofluorscein diacetate ($H_2$DCFDA) with 4T1 cells incubated with $H_2O_2$ as the positive control. As shown in Fig. 5a, the strongest green fluorescence signal as the result of the ROS generation after irradiation was detected in the cells treated with ICG/L-G2R-DA. Quantitative analysis of the amount of laser irradiation-induced ROS

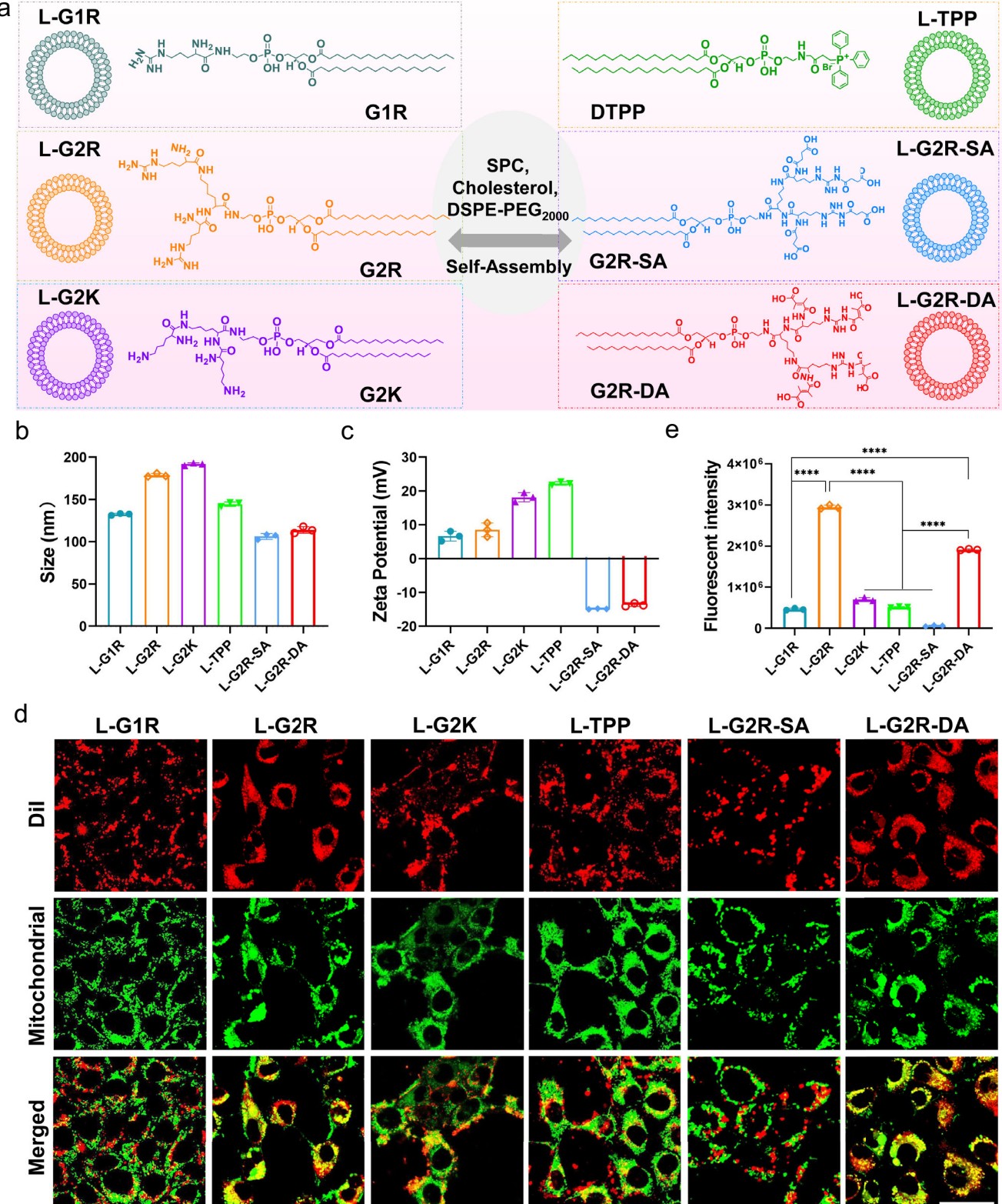

**Fig. 2 Structure characterization and mitochondrial targeting of dendritic lipopeptide-incorporated liposomes. a** Schematic illustration of the structures of L-G1R, L-G2R, L-G2K, L-TPP, L-G2R-SA, and L-G2R-DA. **b** Size distribution of L-G1R, L-G2R, L-G2K, L-TPP, L-G2R-SA, and L-G2R-DA measured by DLS. Data are presented as means ± SEM. ($n = 3$ independent samples). **c** Zeta potentials of L-G1R, L-G2R, L-G2K, L-TPP, L-G2R-SA, and L-G2R-DA in the aqueous solution at pH 7.4. Data are presented as means ± SEM. ($n = 3$ independent samples). **d** CLSM images showing mitochondrial localization of L-G1R, L-G2R, L-G2K, L-TPP, L-G2R-SA, and L-G2R-DA. 4T1 breast cancer cells were incubated with Dil-loaded liposomes for 12 h and then stained with Mitotracker Green FM. Areas with yellow fluorescence in the merged CLSM images denote the co-localization of the liposomes within mitochondria. The scale bar is 40 μm. **e** Flow cytometry analysis of the accumulation of Dil-loaded liposomes in mitochondria of 4T1 cells. Data are presented as means ± SEM. ($n = 3$ biologically independent samples). $P$ values were calculated by the two-tailed Student's $t$-test (****$p < 0.0001$).

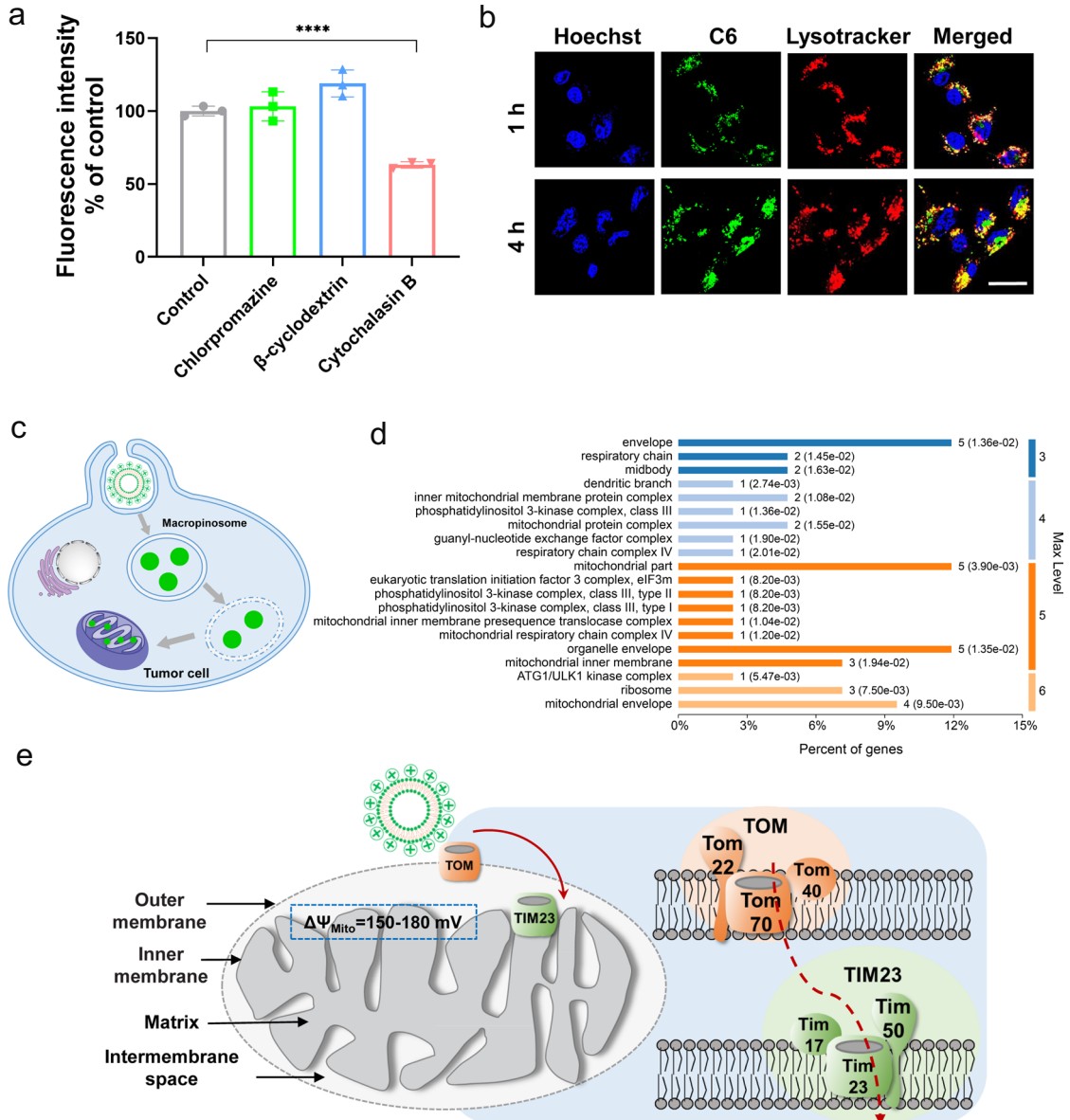

**Fig. 3 Exploration of mitochondrial targeting mechanism. a** The endocytosis inhibition assay on 4T1 cells. Data are presented as means ± SEM. (*n* = 3 biologically independent samples). **b** CLSM images for intracellular tracking of the C6 loaded L-G2R at different time points including C6 channel (green), Lysotracker-stained endosomes channel (red), Hoechst 33342 (blue), and overlay of previous images. The scale bars correspond to 5 μm. **c** The cartoon of the intracellular translocation process. **d** The GeneOntology (GO) pathway analysis according to Cellular Component (CC) for the unique proteins bound on L-G2R compared to liposomes without G2R ingredient. **e** Schematic illustration of mitochondrial targeting mechanism of L-G2R, directly across the TOM and TIM23 machinery into the mitochondrial matrix. *P* values were calculated by the two-tailed Student's *t*-test (****p < 0.0001).

was about 4-fold and 5-fold higher by ICG/L-G2R-DA than those by H₂O₂ and L-TPP (Supplementary Fig. 21).

To investigate the effect of laser irradiation-induced ROS on apoptosis as a result of PDT, the depolarization of the mitochondrial membrane potential (ΔΨm) of cells treated with ICG-loaded liposomes was measured using mitochondrial membrane potential assay kit JC-1 after laser irradiation. In live and undamaged cells with high ΔΨm, JC-1 can accumulate in the mitochondrial matrix in the aggregated form with red fluorescence, while in the apoptotic cells with a lower ΔΨm, we observed JC-1 dispersed in the cytosol emitting green fluorescence. Hence, a low ratio of red to green fluorescence intensities should indicate a decrease in ΔΨm or damage of the mitochondrial membrane. After irradiation, ratios of red to green fluorescence intensities of 4T1 cells incubated with free ICG, ICG/L-TPP, ICG/L-G2R-SA,

and ICG/L-G2R-DA were 37.28 ± 6.57%, 16.44 ± 1.81%, 28.47 ± 0.63%, and 7.23 ± 2.59% (Fig. 5b and c), respectively, indicating that L-G2R-DA caused the most disruption to the mitochondrial membrane.

When quantifying apoptosis of 4T1 cells treated with different ICG-loaded liposomes and laser irradiation using the Annexin V-FITC/PI detection assay and flow cytometry (Fig. 5d), the results showed that the percentages of apoptotic 4T1 cells treated with free ICG, ICG/L-TPP, ICG/L-G2R-SA, ICG/L-G2R-DA were 10.23%, 34.89%, 13.97%, and 55.60%, respectively. The amount of apoptotic 4T1 cells was the highest with ICG/L-G2R-DA treatment.

To further differentiate and confirm the effect of the mitochondrial location of ICG-loaded liposomes, we compared the percentages of apoptotic 4T1 cells and cytotoxicity treated

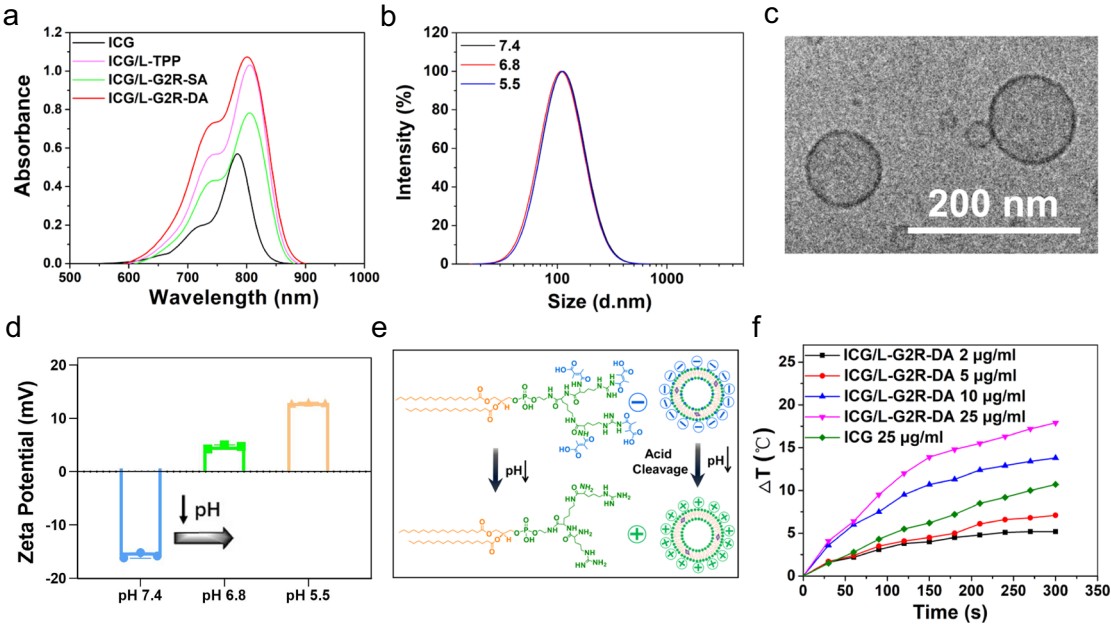

**Fig. 4 Characterization of ICG loaded mitochondrion-specific dendritic lipopeptide liposomes ICG/L-G2R-DA. a** UV/Vis spectra of free ICG, ICG/L-TPP, ICG/L-G2R-SA, and ICG/L-G2R-DA in PBS. **b** Size distribution of ICG/L-G2R-DA in an aqueous solution of pH 7.4, pH 6.8, and 5.5 for 12 h measured by DLS. **c** Cryo-TEM image of ICG/L-G2R-DA. The scale bar is 200 nm. **d** The change of the zeta potential at different pH values. Data are presented as means ± SEM. ($n = 3$ independent samples). **e** Schematic illustration of acid cleavage of ICG/L-G2R-DA. **f** Photothermal temperature curves of ICG/L-G2R-DA at the various concentrations (2–25 μg/mL) and free ICG at high concentration (25 μg/mL) with different exposure times.

with a charge-reversible L-G2R-DA or non-charge-reversible L-G2R-SA carrying ICG and subsequent laser irradiation. As expected, the nearly same level of ICG was observed in 4T1 cells treated with ICG-loaded liposome ICG/L-G2R-SA and ICG/L-G2R-DA (Supplementary Fig. 22a). However, when localized in mitochondria, ICG/L-G2R-DA liposomes showed ~16 folds higher level of ICG in mitochondria than that of ICG/L-G2R-SA liposomes (Supplementary Fig. 22b). Correspondingly, percentages of apoptotic 4T1 cells treated with ICG/L-G2R-SA and ICG/L-G2R-DA were 32.55% and 46.85%, respectively, (Supplementary Fig. 22c). This result was consistent with the findings from the calcein-AM-propidium iodide (PI) double staining for live and dead cells treated with ICG/L-G2R-DA or ICG/L-G2R-SA (Supplementary Fig. 22d), which revealed more rapid and extensive cell death for ICG/L-G2R-DA treatment than for ICG/L-G2R-SA treatment after irradiation. The cytotoxicity induced by irradiating ICG/L-G2R-DA reached 47.85 ± 2.68%, whereas that by ICG/L-G2R-SA was 21.29 ± 1.34% (Supplementary Fig. 22e). Taking together, we conclude that the improved efficacy has resulted from the delivery of ICG to mitochondria of the tumor cell.

**Therapeutic effects of PTT, PDT, and PTT combined with PDT in vitro**. For evaluating the contribution of the PTT effect without the influence of PDT which mainly depends on the ROS produced in photon stimulation, we pre-incubated the cell with ROS inhibiting N-acetylcysteine to quench ROS induced by laser irradiation. Under such conditions, the PTT of 4T1 cells resulted in a cell inhibition rate of 41.5%. For evaluating the contribution of the PDT effect without the influence of the photo-thermal effect, an icebox was used to maintain the temperature of 4T1 cells below 10 °C to eliminate the PTT effect. It was observed that the PDT-induced cell inhibition rate was 49.3. In comparison, the treatment without blocking PTT or PDT induced more than 97% cell death, suggesting a synergistic antitumor effect from the combined PTT and PDT. Worth noting is that treating 4T1 cells with ICG/L-G2R-DA without laser irradiation yielded a negligible cytotoxic effect (Supplementary Fig. 23).

**Antitumor efficacy after combined PTT and PDT in vivo**. 4T1 tumor-bearing mice ($n = 5$/group) were intravenously injected with PBS, ICG, ICG/L-TPP, ICG/L-G2R-SA, and ICG/L-G2R-DA at the dosage of 7.5 mg/kg ICG, respectively, followed by NIR laser irradiation (808 nm, 0.65 W/cm$^2$) for 5 min at 8 h post-injection on day 0 and day 2 with the scheme shown in Fig. 6a. The temperature of the tumor area was monitored by an infrared thermal camera. With NIR irradiation, the average temperatures of the tumor region increased from ~33.5 °C to 43.3 °C, 46.2 °C, and 48.9 °C for the groups treated with free ICG, ICG/L-TPP, and ICG/L-G2R-SA (Fig. 6b), respectively. Interestingly, a much significant increase of the temperature with only 1 min irradiation was observed in the tumors receiving ICG/L-G2R-DA, reaching above 50 °C and then 56.1 °C with 5 min irradiation (Fig. 6b). After intravenously co-injection of different ICG loaded liposome formulations and H$_2$DCFDA, a dye specific to ROS, in mice bearing 4T1 tumors and subsequent NIR laser irradiation for 5 min, the tumor tissue sections from the L-G2R-DA treated group showed the strongest green fluorescence signal from ROS labeled with H$_2$DCFDA (Fig. 6c). The H$_2$DCFDA signal intensity of the ICG/L-G2R-DA treated group was approximately 4.8, 3.2, and 3.1-fold higher than those of ICG, ICG/L-TPP, and ICG/L-G2R-SA groups, respectively (Supplementary Fig. 24). Significantly, we found that tumors treated with ICG/L-G2R-DA completely disappeared with no tumor recurrence observed before being sacrificed after 22 days of treatment (Fig. 6d). In contrast, the mice that received ICG/L-TPP, ICG/L-G2R-SA followed by the same level of NIR laser irradiation exhibited only the marginal tumor inhibition effect comparing to the group that received PBS. No statistically significant weight loss was observed in mice treated with ICG, ICG/L-TPP, ICG/L-G2R-SA, and ICG/L-G2R-DA showed (Fig. 6e).

**Tumor accumulation and blood half time of ICG/L-G2R-DA**. We further examined the tumor accumulation and biodistribution of different ICG-loaded liposomes in the mice bearing

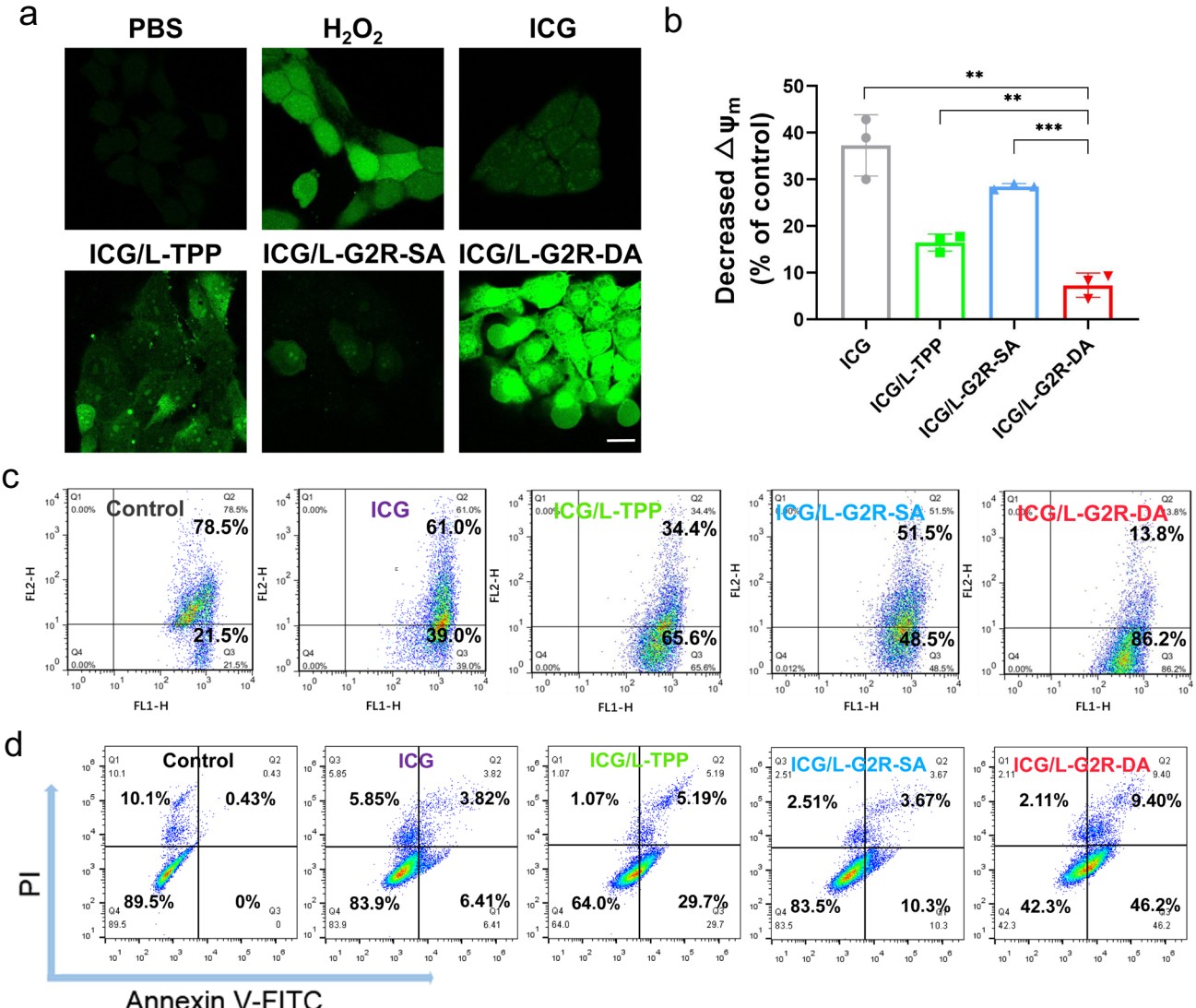

**Fig. 5 4T1 cancer cell apoptosis induced by ICG loaded liposomes. a** CLSM analysis of ROS generation following incubation of the cells with PBS, $H_2O_2$, various ICG containing liposome formulations, and 808 nm laser irradiation. The scale bar represents 50 μm. **b** Change in mitochondrial membrane potential ($\Delta\Psi$m) and **c** Flow cytometry analysis of the relative fluorescent intensity of JC-1 aggregates and monomer of 4T1 cells treated with culture medium, ICG containing liposome formulations for 12 h and NIR laser irradiation. Data are presented as means ± SEM. ($n = 3$ biologically independent samples). **d** Flow cytometry analysis of 4T1 cancer cell apoptosis after treated with the culture medium and various ICG containing liposome formulations for 12 h and NIR laser irradiation. $P$ values were calculated by the two-tailed Student's $t$-test (**$p = 0.0018$, **$p = 0.0076$, ***$p = 0.0002$).

4T1 xenograft tumors based on the measurement of fluorescent intensity from ICG. At 24 h after the injection, ICG/L-G2R-DA exhibited 4.9, 3.0, and 1.6-fold increases in the tumor accumulations comparing to those of free ICG, ICG/L-TPP, and ICG/L-G2R-SA, respectively (Fig. 7a and Supplementary Figs. 25–27). The amount of ICG in the tumor tissue was the highest in all time points in the group treated with ICG/L-G2R-DA compared to other groups. Worth noting, the mice receiving ICG/L-TPP showed the highest accumulation in the liver among all groups, which is likely due to its cationic charge that is known to promote the non-specific accumulation of exogenous materials in the reticular endothelial system (RES) organs, such as liver[16,33].

When we analyzed the pharmacokinetic profiles of all agents tested (Fig. 7a and Supplementary Figs. 25–27), ICG/L-G2R-DA had a longer blood circulation half-life ($t_{1/2}$) at 2 h compared to 0.26 h, 1.13 h, and 1.62 h for free ICG, ICG/L-TPP, and ICG/L-G2R-SA. The area under the curve ($AUC_{0 \sim \infty}$) of ICG/L-G2R-DA

was approximately 5.09, 3.06, and 1.35-fold higher than those of free ICG, ICG/L-TPP, and ICG/L-G2R-SA, respectively.

**Histo-pathological validations**. The mitochondrial accumulation of L-G2R-DA in tumors was examined using immunohistochemistry (IHC) analysis of the frozen tumor tissue sections. With L-G2R-DA labeled with a red fluorescence dye, DiI, and mitochondria stained with MitoTracker Green FM, the signal intensity of L-G2R-DA in mitochondria of tumor cells was about 1.5-fold higher than that of L-TPP and L-G2R-SA labeled with DiI (Fig. 7b and Supplementary Fig. 28). H&E stained tumor tissue collected at 16 h post-treatment showed that necrotic area in the tumors was the largest for the ICG/L-G2R-DA treated group compared to those received ICG, ICG/L-TPP, ICG/L-G2R-SA treatment (Fig. 7c). Measurement of ICG contents in mitochondria isolated from the tumors treated with different agents revealed that the amount of ICG in the mitochondria of tumors treated with L-G2R-

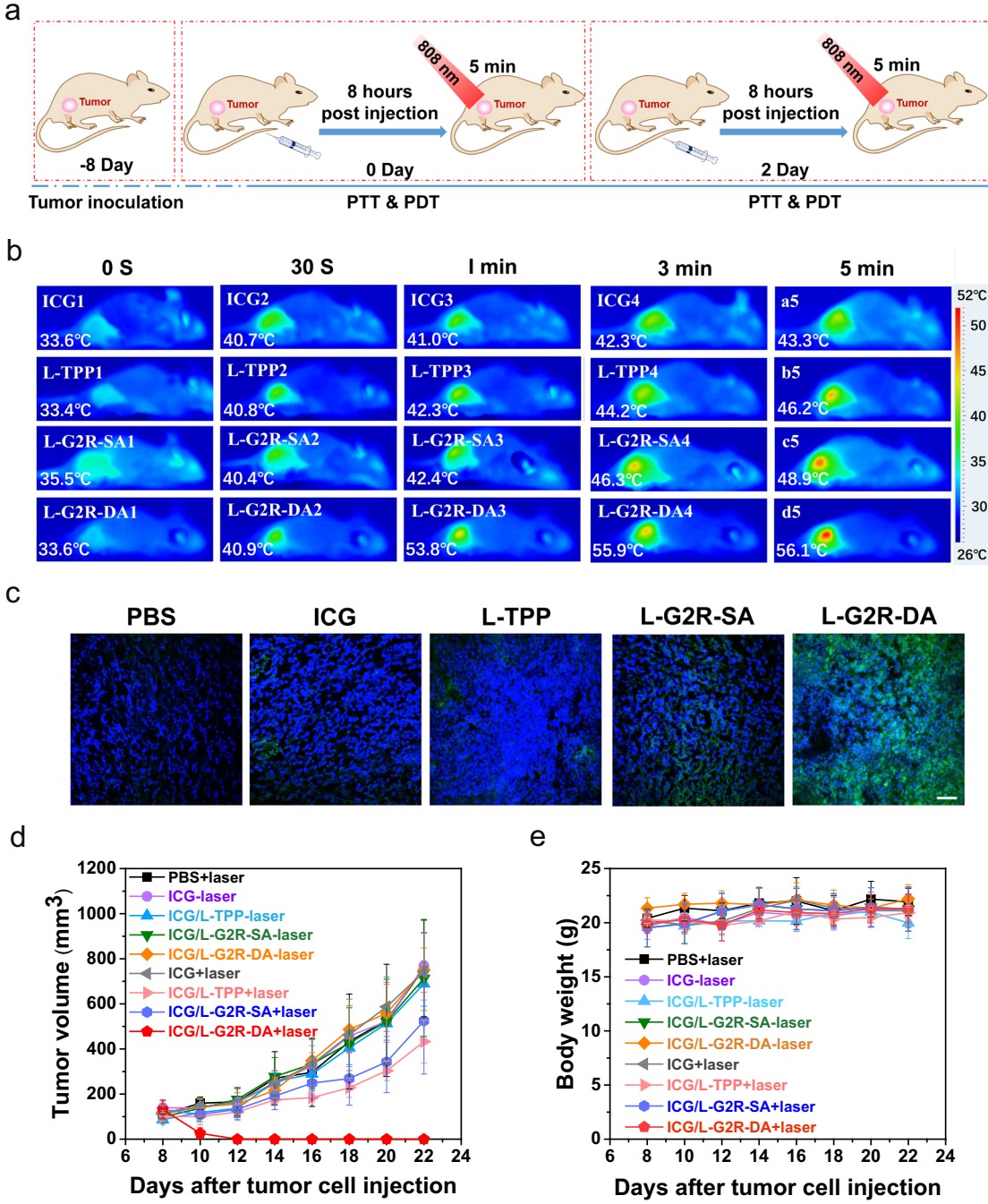

**Fig. 6 Therapeutic efficacy after 4T1 tumor-bearing mice treated with different ICG loaded liposome formulations. a** Schematic illustration of the tumor treatment. **b** Thermographic images and tumor temperature changes of 4T1 tumor-bearing mice at different time points under NIR laser irradiation 8 hours after injection of ICG, L-TPP, L-G2R-SA, and L-G2R-DA. **c** Levels of ROS (green fluorescence) detected in the tumor sections after NIR laser irradiation of the tumors in mice treated with PBS, ICG, ICG/L-TPP, ICG/L-G2R-SA, and ICG/L-G2R-DA. ROS was stained by $H_2DCFDA$ with green fluorescence and nuclei was stained by DAPI with blue fluorescence. The scale bar is 50 μm. **d** Changes of tumor volumes in mice treated with PBS, ICG, ICG/L-TPP, ICG/L-G2R-SA, and ICG/L-G2R-DA with or without NIR laser irradiation. **e** Change of body weights of mice receiving different treatments. **d–e** Data are presented as means ± SEM. ($n = 5$ biologically independent samples).

DA was significantly higher than those treated with L-TPP and L-G2R-SA (Supplementary Fig. 29), further indicating superior mitochondrion targeting specificity in L-G2R-DA.

## Discussion

The mitochondrion is increasingly recognized as an important target for the treatment of many diseases[34–36]. The major challenge for mitochondrion-targeted therapy is the effective delivery of drugs into

the cells and then mitochondria, a sub-cellular compartment. As a classic mitochondrion-targeted moiety, cationic TPPs have widely been used in studying mitochondria biology and developing mitochondrion-targeting therapeutics due to their preferential accumulation in energized mitochondria[37–39]. Covalent linking a TPP moiety to the diagnostic agents, antioxidants, and pharmacophores is a common approach to delivering these agents to mitochondria. While TPP can be delivered to the mitochondrial matrix in the range of 100-

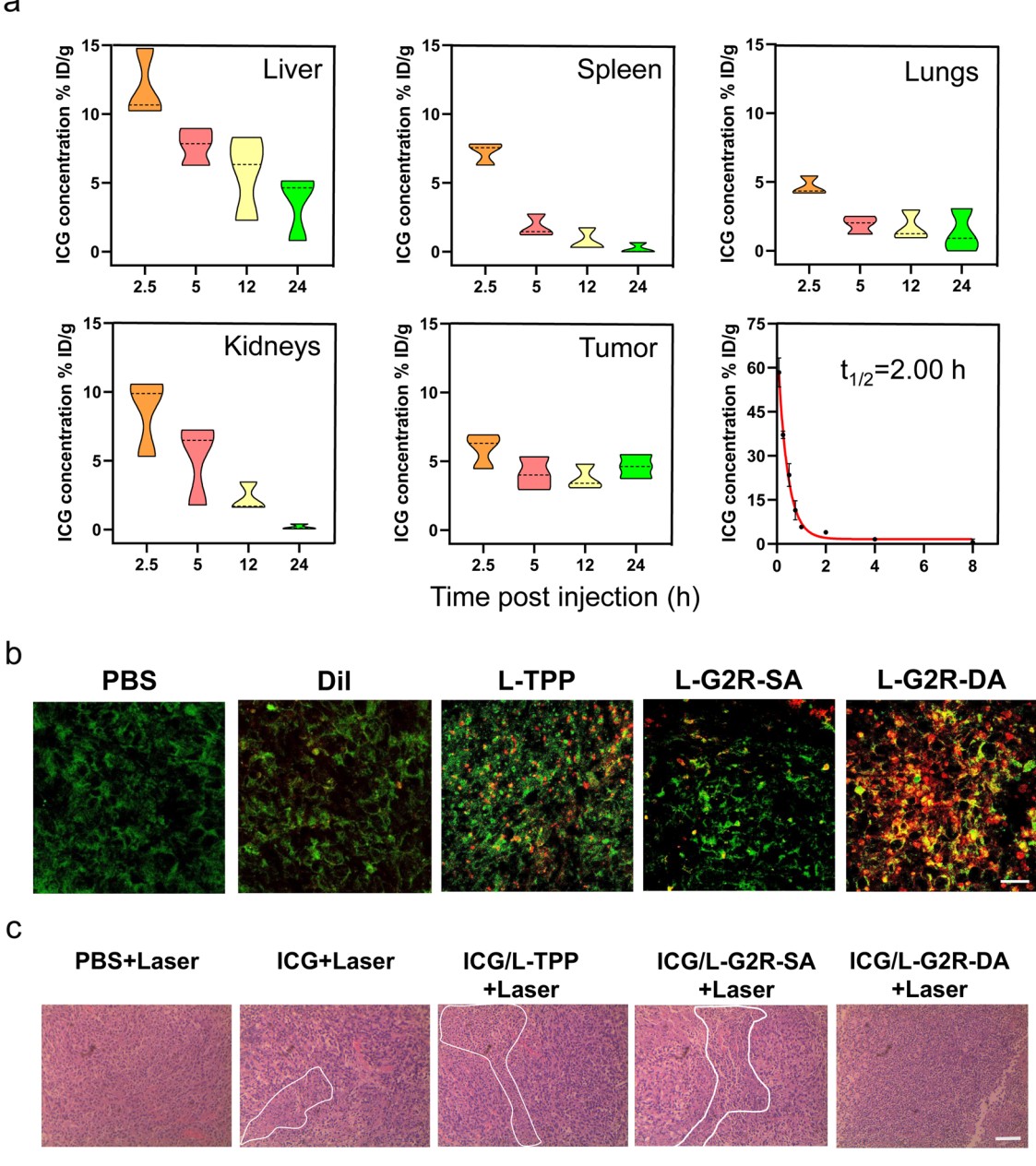

**Fig. 7 Pharmacokinetic profiles of ICG/L-G2R-DA and histological validations. a** The bio-distribution of ICG/L-G2R-DA in different organs. Mice bearing 4T1 tumors were injected intravenously with ICG/L-G2R-DA before sacrificed at different time points. Liver, spleen, lungs, kidneys, tumor, and blood sample were collected to measure the concentration of ICG. The violin plot outlines illustrate kernel probability density, with the area revealing the population of located data. Data are presented as means ± SEM. ($n = 3$ biologically independent samples). **b** Mitochondrial localization of Dil-loaded liposomes observed in the tumor sections. Yellow fluorescence signals in the merged CLSM images denote the co-localization of the liposomes within mitochondria. Red fluorescent Dil from L-G2R-DA and green fluorescence from mitochondria. The scale bar is 20 μm. Experiments were repeated three times independently. **c** H&E staining of tumor tissue collected at 16 h post-PTT and PDT treatment of mice with different treatments. The scale bar represents 250 μm. Experiments were repeated three times independently.

fold to 1000-fold higher concentration than that of the extracellular space[37,40,41], the non-specific accumulation of TPP in the liver and normal tissues due to the cationic charge as observed in this study lead to poor mitochondria-specificity and delivery efficiency with additional concern in normal tissue toxicity. For mimicking natural MPP, using the lipophilic stearoyl tail in addition to the cationic peptide-based dendritic head, we made arginine-rich second-generation dendritic lipopeptide-based liposomes that showed the best mitochondrial targeting efficiency with almost 6-fold higher mitochondrial targeting efficiency than the classic TPP decorated liposomes.

Successful mitochondrion-targeted delivery systems need to overcome multiple biological barriers for sub-cellular delivery upon the tumor accumulation and minimal distribution in normal tissue. We controlled the surface charge of the arginine-rich second-generation dendritic lipopeptide with a pH-responsive component DA, which resulted in prolonged blood circulation and reduced interaction with RES[42,33]. When DA decorated liposomes entered the acidic tumor microenvironment, pH triggered the removal of DA groups restored arginine-rich second-generation dendritic structure. Such arginine-rich molecules mimic cell-

penetrating peptides (CPPs), thus greatly improved cell uptake and intratumor distribution for better therapeutic effect[26,43].

As mitochondrion is also providing a source of intracellular ROS[11,12] that may trigger cell apoptosis, loading FDA approved photosensitizer ICG by L-G2R-DA enables directly deliver ROS inducing PDT[44,45] and hyperthermia agents to the temperature-sensitive mitochondria to deplete the cell "energy supply", and ultimately cell apoptosis. Overall, ICG/L-G2R-DA offers the following benefits in cancer therapy: (1) stealthy surface for drug-loaded liposomes escaping the RES and increasing accumulation in the tumor tissue; (2) pH-responsive surface charge switching to enable sub-cellular mitochondrion-targeting after internalization in tumor cells; and (3) high payload of PDT or PTT agents, such as ICG, delivered to mitochondria.

## Methods

**Synthesis and characterizations of dendritic lipopeptides (DLs).** Amphiphilic dendrimers DLPs with different chemical structures were prepared. The detailed synthetic routes and procedures were described in supporting information. The mass spectroscopy (MALDI-TOF MS, Bruker Autoflex III) and NMR spectrometer ($^1$H NMR, Bruker DRX 400) were used to confirm the structures of synthesized compounds.

**Preparation and characterizations of liposomes.** ICG/L-G2R-DA liposomal vesicles were produced via thin-film hydration. Briefly, trichloromethane: methanol (3:1, V/V) was used to dissolve formulas (SPC: cholesterol: DSPE-PEG$_{2000}$: G2R-DA: ICG = 5: 1: 0.35: 1.2: 1.3, mol: mol) in a round-bottomed flask. A film of dried lipid was obtained after removing the organic solvent and hydrated in pH 7.4 phosphate-buffered saline (PBS) solution at 37 ºC to form phospholipids. After sonication by a probe sonicator, the liposomal vesicles were obtained. The residual free ICG was separated out by a G-25 Sephadex column. Similar procedures were used for preparing the control samples ICG/L-TPP and ICG/L-G2R-SA. The preparation of L-G1R, L-G2R, L-G2K, L-TPP, L-G2R-SA, and L-G2R-DA also used the same method without adding ICG. Encapsulation efficiency (EE) was calculated as EE = $W/W_o \times 100\%$ ($W_0$ was the weight of ICG initially added in the liposome, $W$ was the weight of ICG in the liposomes).

Dynamic light scattering (DLS) analyzer (Brookhaven Zetasizer Nano ZS) was employed to measure the average hydrodynamic size and the zeta potentials of liposomes at different pH. Different liposomes were diluted in the pH 7.4 PBS solution with the phospholipid concentration at 3 mg/ml for measuring the hydrodynamic size. For measuring pH-dependent zeta potential change 0.2 ml of different liposomes were incubated with 2.5 ml PBS (pH 7.4) or acetate buffer (pH 6.8, 5.5) solution at 37 °C, respectively. The two-milliliter solution was taken out at predetermined times to measure the hydrodynamic size and the zeta potentials.

Cryo-TEM (JEM2010, JEOL) was used to observe the morphology of liposomes. Cryo-TEM samples were prepared by dropping liposome solution on the TEM grid and then plunged rapidly into liquid nitrogen. The samples were imaged at about −170 °C and no external staining was employed.

0.5 ml ICG/L-G2R-DA with a series of ICG concentrations (2–25 μg/ml) were taken and stored in ampoules, and then irradiated by 808 nm laser (0.65 W/cm$^2$), separately. Meanwhile, a thermometer was used to measure the temperature of the solution within 300 s. Free ICG at a high concentration of 25 μg/ml was chosen as the control.

**Evaluation of mitochondrion uptake and targeting.** Localization of Dil-loaded liposomes in the mitochondria of 4T1 cells was investigated by a laser confocal scanning microscope (CLSM, LSM 710, Zeiss). 4T1 cells were seeded on a glass-bottomed dish and cultured for 24 h. Various liposomes loaded with fluorescence dye Dil (1 mM Dil) were added and the media cultured for 12 h. For CLSM observation treated 4T1 cells were stained with MitoTracker Green FM (KeyGEN, Nanjing, China) against Dil-loaded liposomes (red). Flow cytometry (Accuri C6, BD Biosciences, USA) was used to measure the drug concentration in the mitochondria. 4T1 cells were seeded on 6-well plates. The different Dil-loaded liposomes (1 mM Dil) were added to the cell containing medium and co-cultured for 12 h. After collecting the cells, mitochondrial isolation was performed using the mitochondria isolation kit (Beyotime Institute of Biotechnology, China). Drug accumulation in mitochondria was estimated based on the measurement of Dil fluorescent intensities in different samples.

All the protocols for the animal tests have been reviewed and approved by the Committee on Animals at Nanjing University and performed in accordance with the guidelines provided by the National Institute of Animal Care. All animals were bred in the pathogen-free facility with a 12 h light/dark cycle at 20 ± 3 °C and had ad libitum access to food and water. Female BALB/c mice bearing 4T1 cells were intravenously injected PBS, free Dil, and Dil-loaded liposomes (Dil, 250 μg/kg), respectively. The tumors were sectioned at 8 h post-irradiation. Frozen sections of tumor tissues were incubated with 100 nmol/L MitoTracker Green FM at 37 °C for

20 min. Fluorescence images were taken with a CLSM. Image J software was used to calculate a Pearson's coefficient.

Mitochondria of the tumors in free ICG, ICG/L-TPP, ICG/L-G2R-SA, and ICG/L-G2R-DA were isolated and the ICG content was measured. Female BALB/c mice bearing 4T1 tumors were intravenously injected with PBS, free ICG, and ICG-loaded liposomes (ICG, 7.5 mg/kg), respectively. The tumors were sectioned at 8 h post-irradiation. Mitochondrial isolation was carried out using a cell mitochondria isolation kit for tissue (Beyotime Institute of Biotechnology, China) and ICG from the mitochondria were extracted using methanol. The concentration of ICG was measured using the fluorescent spectrometer.

**Exploration of the mitochondrial targeting mechanism.** To determine possible endocytosis pathways of the L-G2R, 4T1 cells were pre-incubated with different agents that block different pathways of endocytosis, e.g., MβCD, caveolin mediated endocytosis inhibitor; cytochalasin B, macropinocytosis inhibitor; chlorpromazine, clathrin-mediated endocytosis inhibitor, for 0.5 h at 37 °C. Subsequently, the Dil loaded L-G2R (1 mM Dil) was co-incubated with the cells for 4 h at 37 °C. The fluorescence intensities of loaded L-G2R in the cells treated with different inhibitors were measured by the flow cytometer to determine the effect of each inhibitor or blocked pathway on the cellular uptake of L-G2R.

To estimate the endosomal escape of the L-G2R, 4T1 cells were seeded in the glass-bottomed dish and cultured for 24 h. After incubated with coumarin (C6) loaded L-G2R for 1 h or 4 h, cells were washed with PBS, and then stained with the mixture of Hoechst 33342 and LysoTracker Red for 10 min for CLSM observation.

To investigate the mitochondrial targeting mechanism, L-G2R with a colloidal Fe$_3$O$_4$ (10 nm in diameter) encapsulated was incubated with 4T1 cells at 37 °C for 12 h, and then cell mitochondria DNA isolation kit (Beyotime Institute of Biotechnology, China) was employed for mitochondrial isolation. After magnetic separation and removal of the supernatant, the nonspecifically adsorbed proteins were washed out from Fe$_3$O$_4$ loaded L-G2R. Then, the captured mitochondrial precursor proteins were eluted by loading buffer under powerful shaking for 10 min. The eluate was analyzed by nano-liquid chromatography-tandem mass spectrometry after in-gel digestion with trypsin. Liposomes without G2R ingredients were set as the control group.

L-G2R with gold colloid (10 nm in diameter) encapsulated for visualization of the internalized payload was incubated with seeded 4T1 cells at 37 °C for 12 h, and then the cells were fixed and observed by TEM.

**Measurement of ROS.** After 4T1 cells were seeded in 6-well plates and cultured, they were co-incubated with free ICG, ICG/L-TPP, ICG/L-G2R-SA, and ICG/L-G2R-DA for 12 h. Then 200 μl H$_2$O$_2$ (50 mM) was added into each sample for 30 min. Fluorescent dye H$_2$DCFDA (10 μM) was used to treat all cells for 30 min. The cells were washed twice before being irradiated by 808 nm laser (0.65 W/cm$^2$) for 3 min. ROS in the cells were then detected from H$_2$DCFDA fluorescence using CLSM (LSM 710, Zeiss, Germany) and flow cytometry (Accuri C6, BD Biosciences, USA).

Female BALB/c mice bearing 4T1 cells were intravenously injected fluorescent dye H$_2$DCFDA (10 mM). Then, free ICG, ICG/L-TPP, ICG/L-G2R-SA, and ICG/L-G2R-DA (7.5 mg/kg ICG) were intravenously injected into mice and used 808 nm laser to irradiate after 8 h injection (0.65 W/cm$^2$, 5 min). Frozen sections of tumor tissues were taken fluorescence images by CLSM.

**Mitochondrial membrane potential.** 4T1 cells were seeded on 6-well plates. Free ICG, ICG/L-TPP, ICG/L-G2R-SA, and ICG/L-G2R-DA were cultured with cells for 12 h. Then, cells were washed twice and irradiated by 808 nm laser (0.65 W/cm$^2$) for 3 min. After staining with JC-1 (Beyotime Institute of Biotechnology, China) for 30 min, the cells were harvest and measured for flow cytometer analysis.

**Cell apoptosis.** 4T1 cells were seeded and cultured on a 6-well plate. The cells were cultured with free ICG, ICG/L-TPP, ICG/L-G2R-SA, and ICG/L-G2R-DA for 12 h. After then, cells were irradiated by 808 nm laser for 3 min (0.65 W/cm$^2$) and incubated for an additional 12 h. After washing and centrifugation, the Annexin V-FITC/PI Apoptosis Detection (KeyGEN, Nanjing, China) was employed by flow cytometry in accordance with the specification of the kit.

**Evaluation of PTT, PDT, and PTT&PDT therapeutic effect in vitro.** The relative cell viability under PTT, PDT, and PTT&PDT was evaluated by MTT assay. 4T1 cells were seeded and cultured on 96-well plates. The L-G2R-DA was added into cells. For PTT, N-acetylcysteine was added into 4T1 cells to quench the ROS induced by laser irradiation (808 nm, 0.65 W/cm$^2$, 5 min). For PDT, 4T1 cells were placed on an icebox and subjected to laser irradiation (808 nm, 0.65 W/cm$^2$, 5 min). For PTT combined with PDT, 4T1 cells were treated with 5 min laser irradiation (808 nm, 0.65 W/cm$^2$). After irradiation, cell viability was determined by MTT assay.

**Measurement of blood half time.** Female BALB/c mice were intravenously injected free ICG, ICG/L-TPP, ICG/L-G2R-SA, and ICG/L-G2R-DA (7.5 mg/kg ICG) (n = 3). The blood samples were collected at 0.083, 0.25, 0.5, 0.75, 1, 2, 4, 8 h

post-intravenous injection. Then the blood samples were centrifuged for 8 min at 19,000×g to collect plasma. The plasma samples were mixed with lysis buffer and the ICG concentration was measured by fluorescence spectrometer.

**Biodistribution**. Female BALB/c mice bearing the 4T1 tumors were intravenously injected with free ICG, ICG/L-TPP, ICG/L-G2R-SA, and ICG/L-G2R-DA (7.5 mg/kg ICG), and the main organs, i.e., liver, spleen, heart, lung, and kidney were collected from the mice after sacrificing animals ($n = 3$) at 2.5, 5, 12, and 24 h post-intravenous injection. Then, tissues of collected organs were homogenized in the solvent mixed with saline and methanol (1: 2, V: V). After separating the soluble fractions of the samples by centrifuging, the fluorescence of ICG in the supernatant was measured. The percentage of the injected dose per gram of tissue (% ID/g) was calculated to describe the results of the distribution of each agent.

**Evaluation of the treatment efficacies by PTT, PDT, and combined PTT and PDT**. After intravenous injection of free ICG, ICG/L-TPP, ICG/L-G2R-SA, and ICG/L-G2R-DA (7.5 mg/kg ICG) to mice bearing the 4T1 cells ($n = 3$), 808 nm laser was exposed on tumor after 8 h injection (0.65 W/cm², 5 min). An infrared thermographic camera (Fotric 225, USA) was employed to record the temperature rise of tumor induced by irradiation. AnalyzIR software used to analyze the images.

Female BALB/c mice bearing 4T1 tumors after 8 days of inoculation of 4T1 cells were randomly divided into nine groups ($n = 5$). PBS, ICG, ICG/L-TPP, ICG/L-G2R-SA, and ICG/L-G2R-DA were intravenously injected (ICG, 7.5 mg/kg) on day 0 and 2. Subsequently, laser irradiation for 5 min (808 nm, 0.65 W/cm²) after 8 h post-injection. Tumor volumes ($V$) were calculated as $V = L \times W^2 / 2$ (mm³), where $W$ and $L$ are the larger and smaller diameters of the tumor, respectively. Bodyweight and tumor volume were recorded every two days.

**Histological validation**. Mice bearing 4T1 tumors were intravenously injected PBS, free ICG, ICG/L-TPP, ICG/L-G2R-SA, and ICG/L-G2R-DA (ICG, 7.5 mg/kg), respectively. The tumors were irradiated by 808 nm laser (5 min, 0.65 W/cm²) at 8 h post-injection and sectioned. H&E staining was performed and observed using bright-field microscopy.

**Statistical analysis**. All data were representative results from at least three independent experiments and means ± SEM. The correlation and comparison analyses were performed using the Student's t-test. *$p < 0.05$ was considered statistically significant, **$p < 0.01$, ***$p < 0.001$, and ****$p < 0.0001$ were considered highly statistically significant.

**Reporting summary**. Further information on research design is available in the Nature Research Reporting Summary linked to this article.

## Data availability
The authors declare that the data supporting the findings of this study are available in the article and associated Supplementary Information. Extra data or information are available from the corresponding authors upon reasonable request. Source data are provided with this paper.

Published online: xx xxx 2021

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

## Acknowledgements

This work was supported by the National Natural Science Foundation of China (No. 81601594, 51690153, 21720102005), the National Key R&D Program of China (grant numbers 2017YFA0205400, 2017YFA0701301), and the Natural Science Foundation of Jiangsu Province (BK20202002).

## Author contributions

X.J. and L.J. developed the concepts and designed the experiments; L.J. performed the experiments, measured and analyzed data; S.Z. performed the MTT and ROS detection in vivo experiments; L.J, S.Z., X.Z., C.L, S.J., H.M., and X.J. analyzed results, wrote the and revised the manuscript. X.J. supervised the whole project.

## Competing interests

The authors declare no competing interests.
