## [Peer Review File · Nature Communications]

Reviewers' Comments:

Reviewer #1:

Remarks to the Author:

This work reports a charge-conversion liposome developed to deliver ICG to the mitochondrion (MITO) to induce PDT. It is another report of drug delivery for cancer PDT/PTT, which always showed to "eradicate" tumors. The compounds were carefully characterized.

1. There are so many PDT/PTT systems that just delivered dyes to tumors but were shown to "eradicate" tumors. Is it indeed necessary to deliver the dye to MITO? What is the advantage of this system over those?
2. Was the eradication of the tumor really due to the delivery of ICG to tumor cell MITOs or only due to the high dye concentration? To conclude that the improved efficacy was resulted from the delivery of ICG to tumor cell MITOs, the system should be compared with a control that delivers the same level of ICG to the cancer cell cytosols.
2. Histopathological analysis was very preliminary. MITOs of the tumors in these groups should be isolated and analyzed for a solid conclusion that ICG was indeed delivered to the tumor MITOs.
3. Suppl. Fig 13: L-G2K-DA should be added to compare with L-G2R-DA; furthermore, TPP is known to be very efficient in MITO targeting, but here it was almost not MITO targeting at all; its MITO targeting efficiency was even as low as the negatively charged L-G2R-SA, which is supposed to not MITO localization due to its non-charge conversion(Suppl. Fig 16). This discrepancy leads to the doubt of the data in Suppl. Figs 17&19.
4. Why ICG/L-G2R-DA had longer t_{0.5} and higher AUC than others? particularly ICG/L-G2R-SA? The two systems were so similar. What was the reason that all the liposomes had so short t_{0.5} values even though they all contained PEG on the surface?
5. What was the MITO targeting mechanism? What was the process of intracellular translocation to MITOs? Was it on the outer surface or inside of MITOs?

Reviewer #3:

Remarks to the Author:

The manuscript "Mitochondrion-specific dendritic lipopeptide liposomes for targeted sub-cellular delivery" aims to develop a new liposome-based drug delivery system using dendritic peptide as a mitochondria targeting moiety. This synthesis/formulation is novel and would be interested to researchers in the field of mitochondria targeting for sure. The manuscript is well-written and easy to follow. Experiments are well planned. Questions to the researchers and some suggestions as follow:

1. Could you please explain the logic behind using 4T1 tumor cells? Why did you use this model?
2. In theory, it is likely that pH-sensitive and charge-reversible functions would increase tumor accumulation. However, according to the manuscript, L-G2R had better performance in the mitochondria uptake than L-G2R-DA (higher col-localization). Is there any evidence suggested that using L-G2R-DA is better than L-G2R (in vitro or in vivo)?
3. Line 156-157, I disagree with the authors regarding the vesicular bilayer structure. TEM images are not clear. The bilayer which is an important characteristic of liposomes cannot be detected.
4. Line 157, ICG/L-TPP had a diameter equal to 139 nm which considered much larger than ICG/L-G2RDA and ICG/L-G2R-SA (106 nm and 108 nm, respectively). Could this be one of the reasons why ICG/L-TPP had lower performance?
5. Measuring particle charge (zeta potential) can be tricky. Zeta potential changes with salt concentration. How does the author ensure the reliability of the results obtained in Figure 3e?

Minor suggestions:

6. Figure 6 and Supplementary figure 20-22 would be easier to compare if the same Y-axis (%ID/g) is used.
7. Please specify n in every experiment includes in vitro and in vivo. Do those values represent mean \pm SD or mean \pm SEM?

Response to Reviewer Comments

First of all, we thank the reviewers for the valuable comments and suggestions. The response for reviewers' comments is as following:

To reviewer #1 (Remarks to the Author):

Q1. There are so many PDT/PTT systems that just delivered dyes to tumors but were shown to “eradicate” tumors. Is it indeed necessary to deliver the dye to MITO? What is the advantage of this system over those?

A1. Thanks for reviewer’s good comment. MITO is an important target for many drugs. It is also extremely sensitive to heat shock, which is helpful to improve the effect of PTT^{R1,2}; On the other hand, because singlet oxygen is highly reactive and readily quenched (its lifetime is of the order of 40 ns and its diffusion radius is ~20 nm), its site of production will almost certainly be its site of oxidative damage^{R3}. Thus, MITO may be the most suitable organelle for PDT because it has rich oxygen to produce ROS and even low level of singlet oxygen produced in the mitochondria has larger toxic than that of large amounts produced in other organelle^{R4,5,6}. For example, the injection dose of FDA-approved indocyanine green (ICG) for PDT or PTT is generally 10 mg/kg for tumor-bearing mice, and the laser irradiation power is 1-2 W cm⁻². Although this irradiation power is high enough and may damage adjacent normal tissue, the tumor eradication does not achieve even use of nanoparticulate formulations^{R7,8,9}. Thus, improving treatment efficiency of ICG for PDT and PTT is highly desirable and practicable. In this work, we achieved tumor eradication by used simple dendritic lipopeptide liposomes to deliver ICG to mitochondria with a lower injection dose (7.5 mg/kg) and a lower power (0.65 W cm⁻²). Importantly, the high mitochondrion-targeting of our system can also be useful to those drugs which need to target mitochondrion to exert their therapeutics, especially, to inhibitors of mitochondrial transcription (IMTs).

Q2. Was the eradication of the tumor really due to the delivery of ICG to tumor cell MITOs or only due to the high dye concentration? To conclude that the improved efficacy was resulted from the delivery of ICG to tumor cell MITOs, the system should be compared with a control that delivers the same level of ICG to the cancer cell cytosols.

A2. We thank the reviewer for this important comment and suggestion. To address this suggestion, we have performed additional experiments to compare the effect of cell compartment specific delivery of ICGs using two types of liposomes, a charge-reversible L-G2R-DA, which is mitochondrion-targeting, and non-charge-reversible L-G2R-SA, which isn't mitochondrion-targeting. Two types of liposomes all encapsulated ICG. Firstly, ICG-loaded liposome ICG/L-G2R-SA and ICG/L-G2R-DA were incubated with 4T1 cells at pH 7.4 and 37 °C for 2 h. After cell pellets were collected and washed twice with cold PBS, isolation of mitochondria was carried out using a cell mitochondria isolation kit. ICG from the mitochondria were then extracted using methanol. The concentrations of ICG were measured using fluorescent spectrometer. The results showed that nearly same level of ICG presented in cancer cell for both ICG/L-G2R-SA and ICG/L-G2R-DA samples (**Supplementary Figure 22a**), but ICG in mitochondrial of ICG/L-G2R-DA treated cells showed ~16 folds higher than that of ICG/L-G2R-SA treated cells (**Supplementary Figure 22b**). Subsequently, we investigated cell apoptosis of ICG/L-G2R-SA or ICG/L-G2R-DA-internalized 4T1 cells after cells were irradiated by 808 nm laser for 1 min (0.65 W/cm^2) and incubated for an additional 12 h. After washing and centrifugation of the treated cells, the Annexin V-FITC/PI Apoptosis Detection (KeyGEN, Nanjing, China) was used to measure the apoptotic cells using flow cytometry in accordance with the instructions. The results showed that the percentages of apoptotic 4T1 cells after treatment with ICG/L-G2R-SA and ICG/L-G2R-DA were 32.5% and 46.9%, respectively (**Supplementary Figure 22c**), indicating higher apoptosis occurrence after ICG/L-G2R-DA treatment than that of

ICG/L-G2R-SA. To further support this conclusion, we then investigated cytotoxicity of ICG/L-G2R-SA or ICG/L-G2R-DA-internalized 4T1 cells upon NIR irradiation. The cells were irradiated by 808 nm laser for 3 min (0.65 W/cm^2) and incubated for an additional 12 h. After washing and centrifugation, the propidium iodide (PI) and calcein AM Live/Dead Detection (KeyGEN, Nanjing, China) were employed for fluorescence spectrometer measurement and observation in accordance with the specification of kit. The fluorescence microscopic results in **Supplementary Figure 22d** revealed more rapid and extensive cell death for ICG/L-G2R-DA than for ICG/L-G2R-SA. The cytotoxicity of ICG/L-G2R-DA induced by irradiated reached $47.9 \pm 2.7\%$, whereas this value for ICG/L-G2R-SA was $21.3 \pm 1.3\%$ under similar experimental conditions (**Supplementary Figure 22e**). This result was consistent with the findings from the apoptotic cells. Taking together, we conclude that the improved efficacy was resulted from the delivery of ICG to tumor cell mitochondria. Those results have been added as Supplementary Figure 22 and discussed in the revision.

Supplementary Figure 22. **a** and **b** Fluorescent spectrometer analysis of ICG accumulation in 4T1 cells and their mitochondria of 4T1 cells for both ICG-loaded liposomes, respectively. **c** Flow cytometry analysis of apoptosis of 4T1 cell treated with the culture medium and two ICG-loaded liposomes for 2 h and NIR laser irradiation. **d** CLSM images of 4T1 cells treated with both liposomes following laser irradiation. Live/dead cells are green/red (Calcein AM/PI), respectively. **e** Dead cell ratio (cytotoxicity is defined as a number of PI positive cells of the number of the total cells) of 4T1 cells treated with ICG/L-G2R-SA and ICG/L-G2R-DA following laser irradiation.

Q3. *Histopathological analysis was very preliminary. MITOs of the tumors in these groups should be isolated and analyzed for a solid conclusion that ICG was indeed delivered to the tumor MITOs.*

A3. We thank the reviewer for this suggestion. We have added the new results from the suggested experiments in the revision. Mitochondria of the tumors treated with different agents were isolated from the collected tumor tissues, followed by the extraction and measurement of ICG content using the method described in Answers to your question above (Q2). Mitochondrial isolation was carried out using a cell mitochondria isolation kit for tissue (Beyotime Institute of Biotechnology, China) and ICG from the mitochondria were extracted using methanol. It was noted that the ICG amount of L-G2R-DA in the mitochondria of tumors was approximately increased by 43% and 63% than that of L-TPP and L-G2R-SA, respectively. These results have been added as **Supplementary Figure 29** and summarized in the revision.

Supplementary Figure 29. ICG content in mitochondrial of isolated tumor.

Q4. *Suppl. Fig 13: L-G2K-DA should be added to compare with L-G2R-DA; furthermore, TPP is known to be very efficient in MITO targeting, but here it was almost not MITO targeting at all; its MITO targeting efficiency was even as low as the negatively charged L-G2R-SA, which is supposed to not MITO localization due to its non-charge conversion (Suppl. Fig 16). This discrepancy leads to the doubt of the data in Suppl. Figs 17&19.*

A4. Thanks for the good suggestion and question. In the revision, G2K-DA was synthesized and L-G2K-DA has been added to compare with L-G2R-DA. Flow cytometry was employed to quantify the amount of liposomes in the mitochondria isolated from 4T1 cells. The amount of L-G2R-DA in the mitochondria of 4T1 cells was approximately 3.7-fold higher than that of TPP decorated liposomes, indicating that L-G2R-DA has efficient mitochondrial targeting. As shown in **Supplementary Figure 15**, L-G2R-DA presented approximately 4.3-fold higher accumulation in mitochondria than L-G2K-DA. L-G2R group presented the highest accumulation in mitochondria among all samples. Also, In **Supplementary Figure 13** of previous manuscript, the Pearson's coefficient was calculated on the basis of the degree of co-localization between the red fluorescent and the green fluorescent from the

fluorescence images of Figure 2d. In the revision, we used the flow cytometry to quantify the amount of liposomes in the mitochondria isolated from 4T1 cells to ensure accuracy.

Supplementary Figure 15. a CLSM images showing mitochondrial localization of L-G2K-DA in 4T1 cells. 4T1 breast cancer cells were incubated with Dil-loaded liposomes for 12 h and then stained with Mitotracker Green FM. Areas with yellow fluorescence in the merged CLSM images denote the co-localization of the liposomes within mitochondria. The scale bar is 40 μm . **b** Flow cytometry analysis of accumulation of Dil-loaded liposomes in mitochondria of 4T1 cells. Data are presented as means \pm SD (n = 3). *p < 0.05 and **p < 0.01.

Q5. Why ICG/L-G2R-DA had longer $t_{0.5}$ and higher AUC than others? particularly ICG/L-G2R-SA? The two systems were so similar. What was the reason that all the liposomes had so short $t_{0.5}$ values even though they all contained PEG on the surface?

A5. We agree with reviewer's comment. The systems of ICG/L-G2R-DA and ICG/L-G2R-SA were so similar, which should have similar half-life and AUC. For half-life and AUC, the statistics analyses of ICG/L-G2R-SA group and ICG/L-G2R-DA group were performed using the student t test, the result showed that $p > 0.05$, which was considered no statistically significant. In our research, $t_{1/2}$ ICG = 0.26 h, $t_{1/2}$ ICG/L-TPP = 1.13 h, $t_{1/2}$ ICG/L-G2R-SA = 1.62, $t_{1/2}$ ICG/L-G2R-DA=2.00 h. Compared to free ICG, other groups contained PEG on the surface were greatly extended the half-life. That's similar to what other studies have found^{R10, 11, 12}.

Q6. What was the MITO targeting mechanism? What was the process of intracellular translocation to MITOs? Was it on the outer surface or inside of MITOs?

A6. This is a very important question. In this revision, we probed the MITO targeting mechanism of the liposomes. To elucidate the mechanism of targeting mitochondria by the reported liposomes, we initially used various blockers of specific cellular endocytosis pathways to interrogate the processes of L-G2R internalization in 4T1 cells. As shown in **Figure 3a**, the cellular uptake of L-G2R decreased remarkably in the presence of cytochalasin D ($p < 0.01$), an inhibitor of macropinocytosis.

As shown in the images from CLSM (**Figure 3b**), the fluorescent signals of lysotracker were merged with L-G2R loaded dye coumarin (C6) after incubation for 1 h, indicating that L-G2R were trapped in the endosomes/lysosomes. 4 h later, expanded distribution of signals from C6-loaded L-G2R showed in the cells was observed, indicating L-G2R could mediated endosome escape for cytoplasmic liberation as illustrated in **Figure 3c**. Further, we measured proteins adsorbed on liposome L-G2R using mass spectrometry and analyzed the bound proteins on liposomes according to Cellular Component (CC) using GeneOntology (GO). Compared to the materials in the control group (liposomes without the G2R component, **Supplementary Figure 16**), those in the L-G2R group adsorbed the unique mitochondrial proteins, e.g., mitochondrial inner membrane presequence translocases complex (TIM 23 complex) (**Figure 3d**). The TIM23 translocase consists of the membrane proteins Tim23, Tim17 and Tim50. In addition, the components of translocase of the outer membrane (TOM complex), Tom 70, Tom 22 and Tom 40, were also found. TOM complex is generally considered as main gate for molecules entering into mitochondria. Thus, our observations suggest that L-G2R may directly across the TOM and TIM23 machinery into the mitochondrial matrix, which is the same as a typically precursor protein with an amino-terminal presequence^{R13, 14} (**Figure 3e**). Importantly, transmission electron microscopy (TEM) showed that L-G2R loaded with colloidal gold was located in mitochondria matrix

(Supplementary Figure 17), providing further evidence that L-G2R is able to deliver payloads into the mitochondria matrix. Thus, a possible mechanism of mitochondrion targeting by L-G2R can be considered as: 1) L-G2R liposomes enter cells through macropinocytosis and then undergo endosome escape to cytoplasm; 2) L-G2R liposomes in cytoplasm are transported into the mitochondrial matrix by the TOM and TIM mediated pathway due to the selective adsorption of mitochondrial membrane presequence translocases compared to liposomes without the G2R component.

Figure 3. Exploration of mitochondrial targeting mechanism. **a** The endocytosis inhibition assay on 4T1 cells. Data are presented as means \pm SD (n = 3). *p < 0.05,

****p < 0.01. b** CLSM images for intracellular tracking of the C6-loaded L-G2R at different time points including C6 channel (green), LysoTracker-stained endosomes channel (red), Hoechst33342 (blue) and overlay of previous images. The scale bars correspond to 5 μ m. **c** The cartoon of intracellular translocation process of L-G2R. **d** The GeneOntology (GO) pathway analysis according to Cellular Component (CC) for the unique proteins bound on L-G2R compared to liposomes without G2R ingredient. **e** Schematic illustration of mitochondrial targeting mechanism of L-G2R, directly across the TOM and TIM23 machinery into the mitochondrial matrix.

Reviewer #2 (Remarks to the Author)

Q1. Could you please explain the logic behind using 4T1 tumor cells? Why did you use this model?

A1. In the present study, 4T1 breast cancer cells were chosen for testing anticancer efficacy of PDT/PTT. The tumor growth and metastatic spread of 4T1 cells in BALB/c mice very closely mimic human breast cancer. 4T1 tumor is an animal model for stage IV human breast cancer. 4T1 is also commonly used models for testing the new approaches and methods of drug delivery. Because it is widely used, it is better for comparing our results with those reported in the literature. Therefore, we use 4T1 tumor cells as model in the present research.

Q2. In theory, it is likely that pH-sensitive and charge-reversible functions would increase tumor accumulation. However, according to the manuscript, L-G2R had better performance in the mitochondria uptake than L-G2R-DA (higher col-localization). Is there any evidence suggested that using L-G2R-DA is better than L-G2R (in vitro or in vivo)?

A2. Thanks for reviewer's comment. Yes, L-G2R had better performance in the mitochondria uptake than L-G2R-DA at cell level (**Supplementary Figure 15**). However, at living body level, positively charged nanoparticles, including cationic modified liposomes, cause severe toxicity, instability and a rapid clearance from the

blood compartment after intravenous injection^{15, 16}, thereby limiting their applications in vivo. Ideally, negative charges on the nanoparticle surface can resist non-specific protein absorption in the circulation and avoid rapid elimination by the reticuloendothelial system (RES) to prolonged blood circulation time^{R17,18,19}. The half-life of ICG/L-G2R and ICG/L-G2R-DA was 0.9 h and 2.0 h, respectively. The negative charges on the liposomes surface does prolong the blood circulation time of liposomes. This result indicates that L-G2R-DA is better than L-G2R for use in vivo.

Figure Q1. Blood samples of ICG/L-G2R were collected to measure the concentration of ICG.

Q3. Line 156-157, I disagree with the authors regarding the vesicular bilayer structure. TEM images are not clear. The bilayer which is an important characteristic of liposomes cannot be detected.

A3. We appreciate the reviewer's comment. We have repeated the TEM measurement. This time, Cryo-TEM was employed to show the morphology of ICG/L-G2R-DA, ICG/L-TPP and ICG/L-G2R-SA. The new images included in the revision demonstrated that above liposomes were spherical nanovesicles and similar to that of lipid-based liposomes^{R20, 21}. The thickness of liposomal bilayer we observed was about 5 nm, which was in accordance with those reported in the previous papers^{R22, 23}. The Cryo-TEM images were shown in **Figure 4c** in the revision and **Supplementary Figure 18**.

Figure Q2. Cryo-TEM image of **a** ICG/L-G2R-DA, **b** ICG/L-TPP, **c** ICG/L-G2R-SA. Scale bars are 200 nm.

Q4. Line 157, ICG/L-TPP had a diameter equal to 139 nm which considered much larger than ICG/L-G2RDA and ICG/L-G2R-SA (106 nm and 108 nm, respectively). Could this be one of the reasons why ICG/L-TPP had lower performance?

A4. In order to verify whether the particle size affects performance of ICG/L-TPP, we prepared liposomes with the similar particle size. Liposomes were extruded through membrane filters with 100-nm pores at 50 °C using a water proof Extruder™ (Northern lipids, Vancouver BC, Canada) 10 times. Thus, all the groups showed the similar particle size ~100 nm (Figure Q3a). The amount of liposomes in the mitochondria isolated from 4T1 cells was quantified by Flow cytometry (Figure Q3b). ICG content in mitochondrial of isolated tumor also been detected (Figure Q3c). These results showed that in the similar particle size, ICG/L-G2R-DA still has the most excellent mitochondria targeting both in vitro and in vivo. Particle size doesn't affect the performance of samples.

Figure Q3. **a** Size distribution of L-TPP, L-G2R-SA and L-G2R-DA measured by DLS. Data are presented as means ± SD (n = 3). **b** Flow cytometry analysis of accumulation of Dil-loaded liposomes in mitochondria of 4T1 cells. Data are presented as means ± SD (n = 3). *p < 0.05 and **p < 0.01. **c** ICG content in

mitochondrial of isolated tumor. Data are presented as means \pm SD (n = 3). *p < 0.05 and **p < 0.01.

Q5. Measuring particle charge (zeta potential) can be tricky. Zeta potential changes with salt concentration. How does the author ensure the reliability of the results obtained in Figure 3e?

A5. We thank the reviewer for this question. Indeed, the zeta potential of the charged macromolecules or nanoparticles is dependent on the salt concentration and widely used in the field of nanomedicine. To ensure the reproducibility and reliability of the measurement, in present work, the instrument was initially calibrated with reference standard materials with known zeta potential. The coefficient of variation (CV) for the mean electrophoretic mobility values from each measurement for a reference material is less than 10 %, indicating that the instrument meets the requirement of ISO 13099. Secondly, the pH dependent zeta potentials of various liposomes were measured by DLS (n= 3). Zeta potential of ICG/L-G2R-DA at pH 7.4 was -15.8 ± 0.5 , Zeta potential at pH 6.8 was 4.65 ± 0.4 and zeta potential at pH 5.5 was 12.90 ± 0.1 . CV obtained in measuring those samples was less than 10 %.

Q6. Figure 6 and Supplementary figure 20-22 would be easier to compare if the same Y-axis (%ID/g) is used.

A6. Thanks for this suggestion. The same type of Y-axis (% ID/g) is now used in the Figure 7 and supplementary Figure 25-27 in the revision.

Q7. Please specify n in every experiment includes in vitro and in vivo. Do those values represent mean \pm SD or mean \pm SEM?

A7. Thanks for the suggestion. n and mean \pm SD in every experiment have been specified in the revised manuscript.

References

R1. Flanagan SW, Moseley PL, Buettner GR. Increased flux of free radicals in cells subjected to

- hyperthermia: detection by electron paramagnetic resonance spin trapping. *FEBS Lett* **431**, (1998).
- R2. Zuo L, Christofi FL, Wright VP, Liu CY, Clanton TL. Intra- and extracellular measurement of reactive oxygen species produced during heat stress in diaphragm muscle. *AJP Cell Physiology* **279**, C1058-1066 (2000).
- R3. Moan J, Berg K. The photodegradation of porphyrins in cells can be used to estimate the lifetime of singlet oxygen. *Photochem Photobiol* **53**, 549-553 (2010).
- R4. Simon HU, Haj-Yehia A, Levi-Schaffer F. Role of reactive oxygen species (ROS) in apoptosis induction. *Apoptosis* **5**, 415-418 (2000).
- R5. Sharma V, Anderson D, Dhawan A. Zinc oxide nanoparticles induce oxidative DNA damage and ROS-triggered mitochondria mediated apoptosis in human liver cells (HepG2). *Apoptosis An International Journal on Programmed Cell Death* **17**, 852 (2012).
- R6. Balaban RS, Nemoto S, Finkel T. Mitochondria, Oxidants, and Aging. *Cell* **120**, 483-495 (2005).
- R7. Vaibhav, *et al.* Micellar formulation of indocyanine green for phototherapy of melanoma. *J Control Release*, (2015).
- R8. Tamai K, Mizushima T, Wu X, Inoue A, Yamamoto H. Photodynamic Therapy Using Indocyanine Green Loaded on Super Carbonate Apatite as Minimally Invasive Cancer Treatment. *Mol Cancer Ther* **17**, molcanther.0788.2017 (2018).
- R9. Yang H, *et al.* Micelles assembled with carbocyanine dyes for theranostic near-infrared fluorescent cancer imaging and photothermal therapy. *Biomaterials* **34**, 9124-9133 (2013).
- R10. Sheng Z, *et al.* Smart Human Serum Albumin-Indocyanine Green Nanoparticles Generated by Programmed Assembly for Dual-Modal Imaging-Guided Cancer Synergistic Phototherapy. *ACS Nano* **8**, 12310-12322 (2014).
- R11. Crosasso P, Ceruti M, Brusa P, Arpicco S, Cattel L. Preparation, characterization and properties of sterically stabilized paclitaxel-containing liposomes. *J Control Release* **63**, 19-30 (2000).
- R12. Northfelt DW, *et al.* Doxorubicin encapsulated in liposomes containing surface-bound polyethylene glycol: pharmacokinetics, tumor localization, and safety in patients with AIDS-related Kaposi's sarcoma. *J Clin Pharmacol* **36**, 55-63 (2013).
- R13. Chacinska A, Koehler CM, Milenkovic D, Lithgow T, Pfanner N. Importing mitochondrial proteins: machineries and mechanisms. *Cell* **138**, 628-644 (2009).
- R14. Pfanner BN. Mechanisms of Protein Import into Mitochondria. *Curr Biol*, (2003).
- R15. Guan J, *et al.* Enhanced immunocompatibility of ligand-targeted liposomes by attenuating natural IgM absorption. *Nature Communications* **9**, 2982- (2018).
- R16. A TJ, *et al.* Dual-functional liposomes based on pH-responsive cell-penetrating peptide and hyaluronic acid for tumor-targeted anticancer drug delivery. *Biomaterials* **33**, 9246-9258 (2012).
- R17. Gratton SEA, *et al.* The effect of particle design on cellular internalization pathways. *Proc Natl Acad Sci U S A* **105**, 11613-11618 (2008).
- R18. Cho EC, Xie J, Wurm PA, Xia Y. Understanding the role of surface charges in cellular adsorption versus internalization by selectively removing gold nanoparticles on the cell

- surface with a I2/KI etchant. *Nano Lett* **9**, 1080 (2009).
- R19. Jiang L, *et al.* Overcoming drug-resistant lung cancer by paclitaxel loaded dual-functional liposomes with mitochondria targeting and pH-response. *Biomaterials* **52**, 126-139 (2015).
- R20. Bollhorst T, Rezwan K, Maas M. Colloidal capsules: Nano- and microcapsules with colloidal particle shells. *Chem Soc Rev* **46**, (2017).
- R21. Fox, *et al.* Cryogenic transmission electron microscopy of recombinant tuberculosis vaccine antigen with anionic liposomes reveals formation of flattened liposomes. *International Journal of Nanomedicine* **9**, 1367-1377 (2014).
- R22. Battersby BJ, Grimm R, Huebner S, Cevc G. Evidence for three-dimensional interlayer correlations in cationic lipid-DNA complexes as observed by cryo-electron microscopy. *Biochim Biophys Acta* **1372**, 379-383 (1998).
- R23. Kotouek J, Hubatka F, Maek J, Kulich P, Turánek J. Preparation of nanoliposomes by microfluidic mixing in herring-bone channel and the role of membrane fluidity in liposomes formation. *Sci Rep* **10**, (2020).

Reviewers' Comments:

Reviewer #1:

Remarks to the Author:

The authors have addressed my concerns carefully with new experiments. So, I recommend acceptance.

Reviewer #3:

Remarks to the Author:

I thank the authors for the clarification and the works that have been done to strengthen the manuscript. The authors have addressed all my concerns.

Response to Reviewer Comments

To reviewer #1 (Remarks to the Author):

Q1. The authors have addressed my concerns carefully with new experiments. So, I recommend acceptance.

A1. Thanks for reviewer's valuable comments and suggestions.

Reviewer #3 (Remarks to the Author)

Q1. I thank the authors for the clarification and the works that have been done to strengthen the manuscript. The authors have addressed all my concerns.

A1. Thanks for reviewer's valuable comments and suggestions.